# *Mycobacterium szulgai* Lung Disease or Breast Cancer Relapse—Case Report

**DOI:** 10.3390/antibiotics9080482

**Published:** 2020-08-05

**Authors:** Anna Kempisty, Ewa Augustynowicz-Kopec, Lucyna Opoka, Monika Szturmowicz

**Affiliations:** 11st Department of Lung Diseases, National Tuberculosis and Lung Diseases Research Institute, 01-138 Warsaw, Poland; monika.szturmowicz@gmail.com; 2Department of Microbiology, National Tuberculosis and Lung Diseases Research Institute, 01-138 Warsaw, Poland; e.kopec@igichp.edu.pl; 3Department of Radiology, National Tuberculosis and Lung Diseases Research Institute, 01-138 Warsaw, Poland; lucyna.opoka@gmail.com

**Keywords:** mycobacteria, breast cancer, *Mycobacterium szulgai*

## Abstract

Cancers are one of the risk factors of non-tuberculous mycobacterial (NTM) lung disease. The majority of data in this group of patients concern infections caused by *Mycobacterium avium*—the most prevalent NTM species worldwide. In contrast, limited information can be found regarding the uncommon NTM such as *Mycobacterium szulgai*. We present the case of *M. szulgai* lung disease in a patient with a history of breast cancer. Coexistence of NTM lung disease and breast cancer lung metastasis as well as primary lung cancer was suspected. Finally, neoplastic disease was ruled out based on negative results of endobronchial biopsy and negative tumor markers for lung and breast cancer. *M. szulgai* lung disease was successfully treated with rifampicin, ethambutol and clarithromycin.

## 1. Introduction

Non-tuberculous mycobacteria (NTM), also known as mycobacteria other than tuberculosis or atypical mycobacteria, are the group of microorganisms that are widespread in the environment. Exposure to NTM can lead to the disease under certain conditions. Of 180 known NTM species only a few are clinically relevant for humans. Moreover, susceptibility to development of NTM diseases is not only determined by the type of pathogen, but also by host-risk factors, such as structural lung defects, genetic defects and immunodeficiencies [1].

The most common manifestation of NTM disease is lung disease. Isolation of NTM from respiratory specimens is not synonymous with NTM lung disease; it may be also caused by colonization or even contamination. So, according to the American Thoracic Society/Infectious Diseases Society of America (ATS/IDSA) statement on nontuberculous mycobacterial disease, a combination of clinical, radiologic and microbiologic criteria must be met to establish a diagnosis of NTM lung disease [2].

Cancers are the risk factors of NTM lung disease development. The most predominant for NTM disease are hematologic malignancies and lung cancer, with *Mycobacterium avium* as the most common causative pathogen [3,4]. On the other hand, radiologic features of NTM lung disease may mimic malignancy [5,6]. A coexistence of NTM infection and cancer in the same lesion has also been reported [7].

*Mycobacterium szulgai* is an uncommon pathogen with a frequency of isolation estimated at less than 1% of all NTM species [8]. Fifty years have passed since the first case report of *M. szulgai* infection; nevertheless, the data on the course of the disease among cancer patients as well as in the general population are very limited [9].

We decided to present a case of *M. szulgai* lung disease in a patient with a history of breast cancer for the reasons mentioned above.

## 2. Case Report

A 64-years female former smoker was admitted to the Department of Lung Diseases due to productive cough, fever and 8 kg weight loss within the last three months. Prior to the hospitalization the patient was unsuccessfully treated with antibiotics. At the age of 48, the patient was diagnosed with breast cancer and underwent total right-sided mastectomy, followed by radiotherapy. Oncologic follow-up did not reveal the relapse.

On admission the patient was afebrile, her body mass index was within normal range (19.2 kg/m^2^) and on physical examination no abnormalities were detected.

Laboratory tests revealed an elevated erythrocyte sedimentation rate (85 mm/h) and C-reactive protein 67 mg/L (<5 mg/L). Full blood count was normal. Tumor markers for lung (Cyfra 21-1, carcinoembryonic antigen) and breast cancer (CA 15-3) were negative.

A chest X-ray (Figure 1a) showed thick-walled cavity in the upper left lobe associated with multiple nodular opacities up to 20 mm in size in both lungs. Patchy consolidations at the base of the right upper lobe were observed. A computed tomography of the chest (CT scan) was consistent with the chest X-ray (Figure 2a).

Bronchoscopy revealed a chronic inflammatory process in bronchi. Submucosal infiltration with reduction in the autofluorescence (AF) intensity and nodules in the left B1 + 2 were seen. The bifurcation between B1 + 2 and B3 was widened. Histologic analysis of endobronchial biopsy showed no carcinoma cells, but only high-grade inflammation. Ziehl–Neelsen staining of the biopsy specimen revealed the presence of mycobacteria. Bronchial washing microscopic smear for acid-fast bacilli was positive. A nucleic acid amplification test (GeneXpert) was performed and the result was negative—no DNA of the *M. tuberculosis* complex was found in the sample. After 10 days, the NTM strain was cultured on liquid media using the BD BACTEC MGIT 960 system. The isolated strain was identified using GenoType Mycobacterium CM/AM assay (HAIN Lifescience, Germany) as *Mycobacterium szulgai* (Figure 3).

All the ATS/IDSA diagnostic criteria for NTM pulmonary disease were achieved [2]. The treatment with rifampicin 600 mg, ethambutol 750 mg and clarithromycin 1000 mg was initiated. Conversion to negative culture was documented after eight months. The treatment was continued for an additional 12 months. Chest X-ray and CT performed at the end of treatment showed a regression of nodular opacities in both lungs (Figure 1b and Figure 2b). The cavity in the left upper lobe was slightly dimished; the wall thickness was reduced. During a one year follow up no signs of breast cancer relapse were observed. A written informed consent for publication has been obtained from the patient.

## 3. Discussion

*M. szulgai*, an uncommon, slow growing NTM, was named in honor of Polish microbiologist Teofil Szulga for his contribution to the development of the lipid analysis used for identification of this species [9]. Epidemiological data on *M. szulgai* are very limited. An NTM Network European Trials Group collaborative study showed that of all NTM isolates from pulmonary samples, *M. szulgai* was identified depending on the geographic region from 0% in Africa to 0.77% in Northern Europe [8]. In Poland, between 2013 and 2017, of 2799 NTM isolates, 12 (0.43%) *M. szulgai* cases were recorded [10].

According to the ATS/IDSA statement, due to its rare occurrence in the environment *M. szulgai* isolation almost always shows pathological relevance [2]. Results from the Netherlands confirmed that almost 80% of the patients in whom *M. szulgai* was isolated fulfilled all criteria to diagnose *M. szulgai* lung disease [11]. In contrast, a study from Korea showed that only 43% of patients with positive respiratory isolates had *M. szulgai* disease [12].

The most common form of *M. szulgai* infection is lung disease, but extrapulmonary localization and even disseminated disease have also been reported [11,12].

Cancers are one of the risk factors associated with NTM disease development. Among cancer patients in Taiwan, NTM disease was diagnosed in 0.15% of cases [3]. The risk was 4.43 times higher compared to non-cancer controls. Interestingly, elevated risk persisted until more than one year after cancer onset. So far, only one case of a patient who developed *M. szulgai* lung disease eight years after lung cancer surgery has been reported [13]. Our case concerned a female treated for breast cancer in the past.

All patients suspected of NTM lung disease require extensive work-up to avoid misdiagnosis. According to the ATS/IDSA statement, the appropriate exclusion of other alternative diagnoses is essential, as well as evaluation of the chest X-ray or chest CT scan [2]. Hong et al. reported 3.6% cases of NTM lung disease mimicking lung cancer [5]. Common radiologic manifestation of *M. szulgai* lung disease is a fibrocavitary form. Nodular opacities in the course of *M. szulgai* infection occur less frequently; nevertheless, both forms can be suspected for malignancy [12]. Opoka et al. reported a rare case of squamous lung cancer manifested by thickening wall of cavity, which was previously stable during a two year follow-up period [14]. In our case, coexistence of NTM infection and breast cancer lung metastasis as well as primary lung cancer had been suspected. Currently, ^18^F-FDG positron emission tomography/computed tomography (^18^F-FDG PET/CT) is considered as the best method for detecting lung metastases with higher sensitivity to conventional imaging [15]. However, studies on the differentiation between benign and malignant lesions using ^18^F-FDG PET/CT have been inconclusive. Figueroa et al. showed that the specificity of ^18^F-FDG PET/CT is not sufficient to differentiate between lung nodules caused by NTM and cancer, as an increased maximum standardized uptake value (SUV max) is obtained in both types of lesion [16]. According to the authors, an SUV max value ≥ 16 may be more specific for cancer diagnosis as such values were not observed in NTM lesions. In contrast, Taralli et al. noted a lower prevalence of malignancy among patients with previous cancer history and multiple pulmonary nodules compared to those with solitary nodules [17]. Despite this fact, they proposed personalized management to avoid misdiagnosis. In our case, cancer was finally ruled out based on the following evidence: Negative result of endobronchial biopsy specimen, negative tumor marker for lung and breast cancer and improvement under the treatment of *M. szulgai* disease.

A treatment schedule for *M. szulgai* is not precisely established in the ATS/IDSA statement [2]. The most commonly used regimen includes rifampicin, ethambutol and macrolides and/or quinolones [11]. Our patient has been effectively treated with rifampicin, ethambutol and clarithromycin. Due to clinical similarity to tuberculosis, a four drugs regimen consisting of isoniazid, rifampicin, ethambutol and pyrazinamide has been occasionally used in other patients for the first two months or until *M. szulgai* identification [18].

## 4. Conclusions

In conclusion, we have presented a case of NTM lung disease caused by the rare pathogen *M. szulgai* in a patient treated for breast cancer in the past. The exclusion of cancer relapse in our patient was a big challenge.

## Figures and Tables

**Figure 1 antibiotics-09-00482-f001:**
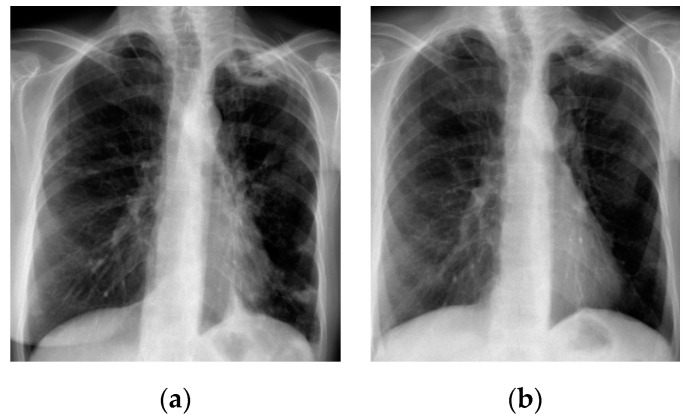
Chest X-ray; (**a**) before treatment—thick-walled cavity in the upper left lobe, multiple nodular opacities in both lungs and patchy consolidation at the base of the right upper lobe; (**b**) after treatment—regression of nodular opacities in both lungs, slight reduction of the cavity in the left upper lobe.

**Figure 2 antibiotics-09-00482-f002:**
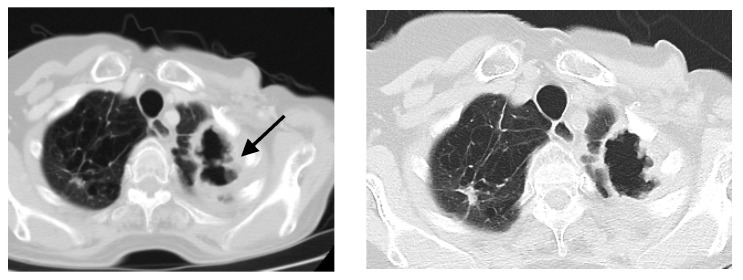
Axial CT scan of a lung window; (**a**) before treatment—large cavity, 40 × 70 mm in size, with irregularly thickened wall (black arrow) and multiple lung nodules up to 20 mm in size in both lungs; (**b**) after treatment—the cavity in the upper left lobe slightly diminished with the reduction of its wall thickness and regression of nodular opacities in both lungs.

**Figure 3 antibiotics-09-00482-f003:**
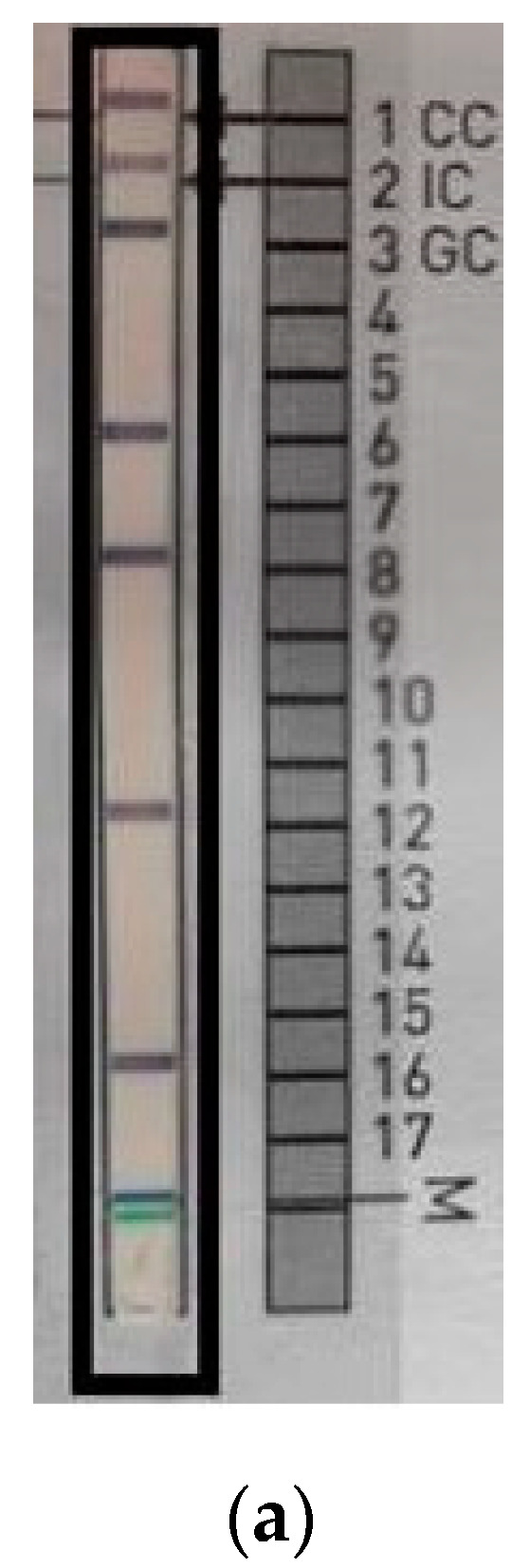
*Mycobacterium szulgai* identification test using the molecular method GenoType Mycobacterium CM/AM assay. Strip with results of *M. szulgai* identification from patient’s sample (**a**); interpretation according to the interpretation chart provided by manufacturer, HAIN Lifescience, Germany (**b**).

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
