# Peer review of "Mycobacterium szulgai Lung Disease or Breast Cancer Relapse—Case Report"

_antibiotics, 2020, doi:10.3390/antibiotics9080482_

Round 1
Reviewer 1 Report
The given MS is the Case report of the single patient, investigating with M. szulgai infection or breast cancer relapse. The Case seems interesting as NTM lung disease is less common. However, MS needs a rigorous understanding of the following factors:
1) The authors mentioned Myco infection identification and confirmation has been done using tests. These test results and confirmation reports need to be presented in the publication for a comprehensive understanding of the case.
2)Authors have not mentioned about the ethical compliance and consent from the patient. Also, case report publication needs to follow CARE guidelines. The author need to mention whether they followed these guidelines?
3) Are any such cases are reports previously in the literature with similar or different ethnicity, needs to be discussed, Along with the strength and weaknesses of the current case report.
Reviewer 2 Report
In this manuscript the authors present the case of M. szulgai lung disease in the patient with history of breast cancer. Although the authors mention that this infection is rare, there are a multitude of cases in the literature and this case does not present anything new or particularly significant from the others.
Major comments:
1. correction of the title: replacement of M. szulgai with Mycobacterium szulgai
2. Figures 1 and 2 require a legend explaining their contents, because not all readers are specialists in radiology.
3. A list of abbreviations in alphabetical order must be added at the end because there are many abbreviations not explained in the text. Example ATS / IDSA, ERS and so on.
4. Discussions are limited only to the association of M. szulgai in cancer patients. To improve the value of the manuscript, discussions should be extended to patients with other immune deficiency diseases.
5. NTM including M. szulgai can be a potential cause of infection even in patients without severe immunosuppression. Discussions should include this aspect as well.
5. What is the strength of this case? What differentiates it from the dozens presented in the literature?
6. The manuscript has no conclusions; they must be added.
Consider revising accordingly.
Round 2
Reviewer 1 Report
The authors have answered the queries raised by the reviewer. Although the current case study involves single patient report, it is an interesting case report.
Reviewer 2 Report
The manuscript is improved. I endorse publication.